# The Seniors’ Community Hub: An Integrated Model of Care for the Identification and Management of Frailty in Primary Care

**DOI:** 10.3390/geriatrics6010018

**Published:** 2021-02-12

**Authors:** Marjan Abbasi, Sheny Khera, Julia Dabravolskaj, Bernadette Chevalier, Kelly Parker

**Affiliations:** 1Department of Family Medicine, University of Alberta, Edmonton, AB T6G 1K4, Canada; sheny.khera@gmail.com (S.K.); b.chevalier@ualberta.ca (B.C.); 2School of Public Health, University of Alberta, Edmonton, AB T6G 1C9, Canada; dabravol@ualberta.ca; 3Edmonton Oliver Primary Care Network, Edmonton, AB T5G 0E5, Canada; KParker@eopcn.ca

**Keywords:** frailty, community-dwelling older adults, primary care, integrated care model

## Abstract

(1) Background: Integrated models of primary care deliver the comprehensive and preventative approach needed to identify and manage frailty in older people. Seniors’ Community Hub (SCH) was developed to deliver person-centered, evidence-informed, coordinated, and integrated care services to older community dwelling adults living with frailty. This paper aims to describe the SCH model, and to present patient-oriented results of the pilot. (2) Methods: SCH was piloted in an academic clinic with six family physicians. Eligible patients were community dwelling, 65 years of age and older, and considered to be at risk of frailty (eFI > 0.12). Health professionals within the clinic received training in geriatrics and interprofessional teamwork to form the SCH team working with family physicians, patients and caregivers. The SCH intervention consisted of a team-based multi-domain assessment with person-centered care planning and follow-up. Patient-oriented outcomes (EQ-5D-5L and EQ-VAS) and 4-metre gait speed were measured at initial visit and 12 months later. (3) Results: 88 patients were enrolled in the pilot from April 2016–December 2018. No statistically significant differences in EQ-5D-5L/VAS or the 4-metre gait speed were detected in 38 patients completing the 12-month assessment. (4) Conclusions: Future larger scale studies of longer duration are needed to demonstrate impacts of integrated models of primary care on patient-oriented outcomes for older adults living with frailty.

## 1. Introduction

Frailty is one of the major challenges facing older adults, described as “the most problematic expression of population aging” [1] (p. 752). While frailty is not necessarily synonymous with aging, it is highly prevalent in older adults. It is estimated that globally 1 in 6 community dwelling older adults may be living with frailty [2]. In Canada, more than 1.5 million older adults are currently living with frailty, and in 10 years this number is projected to increase to two million [3,4]. Seniors living with unidentified and unmanaged frailty often present to health care systems in acute crisis, with frequent emergency room visits, lengthy hospital admissions, decreased quality of life, and substantial health and social care costs [5,6].

Frailty is defined as a state of heightened vulnerability often resulting from age-related changes in the function of multiple body systems. Frailty involves a dynamic interplay of multiple biopsychosocial factors that can leave individuals with a decreased ability to adequately respond to stressors, and puts them at increased risk for poor health outcomes [7]. Individuals experience states of robust health, vulnerability, and mild, moderate or severe frailty, as they transition along the fitness to frailty spectrum over time [6]. Evidence supports that early frailty identification and management can mitigate or slow down these transitions [1]. However, due to its complexity, traditional single-disease focused models of care are not helpful when it comes to frailty identification and management [7]. In fact, the concept of frailty drives a much-needed paradigm shift towards a holistic and integrated care approach [8,9]. Integrated care is defined as “a well-planned and well organized set of services and care processes, targeted at the multi-dimensional needs/problems of an individual client, or a category of people with similar needs/problems” [10] (p. 18).

The World Health Organization Integrated Care for Older People (WHO ICOPE) is a useful guide for the development of integrated care initiatives. According to it, successful integrated care for older people should include: “comprehensive assessments and integrated care plans; shared decision-making and goal setting; support for self-management; multidisciplinary teams; unified information or data-sharing systems; community linkages or integration; and supportive leadership, governance and financing mechanisms.” [11] (p. 4) Importantly, building capacity within multidisciplinary teams is one of the key recommendations of the WHO World Report on Ageing and Health [12].

A systematic review conducted in 2017 outlined other key features of integrated care models for frailty, such as (1) improved targeting of high-risk community-dwelling seniors using validated screening tools to identify people in need for a comprehensive geriatric assessment; (2) promotion of restorative approach to care; (3) promotion of self-management skills; (4) proactive and coordinated care between case managers and family physicians; (5) tailored multidimensional interventions; (6) exploration of new models and ways to deliver comprehensive geriatric assessment; (7) increasing knowledge of frailty and promoting skills and competencies on the subject; (8) support adoption of successful initiatives and assure their implementation in the real setting; and (9) shift towards person-centered outcomes [13].

The essential role in this paradigm shift towards integrated care belongs to primary care, since its person-centered approach has been shown to meet the needs of frail older adults [14,15] and is ideally situated for implementation of integrated care models. However, timely recognition and proactive intervention of at-risk seniors presents many challenges to primary care providers. Therefore, a practical evidence-based guide or model of care is required to support primary care providers in caring for older adults with frailty.

Seniors’ Community Hub (SCH) is an integrated care model that builds capacity within primary care to deliver person-centered, evidence-informed, coordinated, and holistic care services to older adults living with frailty. The SCH model of care is well aligned with recommendations for integrated care models by WHO and others [11,16,17,18,19]. It was first piloted in one academic clinic from 2016–2018 and has evolved and expanded over time. The SCH provides a structural framework that primary care physicians can use and adapt within their own practices by mobilizing available resources in their clinics and community partners to deliver this team-based approach. The SCH was recognized as the top innovation from the Canadian Frailty Network in 2018 [20], and the model showcased nationally as part of the Accelerating Frailty Care in the Community Collaborative (AFCC), Canadian Foundation for Healthcare Improvement. The aim of this paper is to describe the SCH, an integrated primary care model, and to present patient-oriented results of the evaluation of the SCH pilot project.

## 2. Materials and Methods

### 2.1. Description of the Model of Care

The SCH model of care, co-founded by a care of the elderly (COE) physician (MA) and a primary care physician (SK), was implemented in partnership with the Edmonton Oliver Primary Care Network (EOPCN), Alberta, Canada. The EOPCN is one of 41 PCNs within Alberta that are comprised of groups of family physicians and other health care professionals that work together to provide comprehensive primary care services to Albertans.

The SCH was launched as a pilot project at an academic, community-based clinic composed of six family physician practices in Edmonton, Alberta. This clinic is one of EOPCN’s 40 partner physician clinics. A developmental evaluation framework which is a process that supports innovation within evolving programs was used in the implementation of the SCH. Patton describes developmental evaluation frameworks as those that support innovations taking place in environments of complexity and uncertainty, with the primary objective of exploring and experimenting with innovations instead of being focused on finding a specific solution to the problem [21]. These principles of developmental evaluation were applied in the SCH implementation permitting iterative changes to be made based on real-time feedback received from the clinical team, patients and caregivers. Recognizing that change is difficult, and some team members could perceive implementing changes in clinic practices as disruptive, strategies to facilitate change management were incorporated throughout the implementation. The change management strategy, ADKAR (Awareness-Desire-Knowledge-Ability-Reinforcement), proved to be helpful in guiding organizational change and implementing the SCH [22]. Examples of how these ADKAR strategies were applied will be discussed in the context of the Lessons learned in the Discussion section.

The overarching goal of the SCH was to transform primary care into a central hub to better meet the health and social needs of older adults and their caregivers. Key features of the SCH include: a structured process of care for identifying, assessing and managing frailty; education of the healthcare workforce in the principles of geriatrics and effective team-based care delivery; patient and caregiver empowerment with direct input on the design and delivery of SCH care services; partnerships in care to enable integration of medical and social/community services; and an evaluation framework to foster consistency in care and long-term sustainability.

### 2.2. Patient Recruitment for SCH Pilot

Patients of the participating clinic of six family physicians’ practices, who were 65 years of age and older, community dwelling, had an electronic frailty index (eFI) score of >0.12, and provided consent were included in the SCH pilot project. Patients were excluded if they were under 65 years of age, did not provide consent to participate in the SCH, eFI (0–0.12), or were living in a long-term care facility at the time of enrollment.

Recruitment of patients occurred by multiple routes. Patients and caregivers were made aware of the SCH with posters and brochures with information about the program displayed in the clinic, and were able to self-refer by speaking to any team member at this clinic. Primary health care providers recruited patients during routine clinic visits with any of the following warning features that may indicate risk of frailty: multiple comorbidities (2 or more chronic diseases), polypharmacy (5 or more chronic medications), unintentional weight loss, fatigue, chronic pain, changes in functional status, depression, anxiety, changes in memory or personality, increased alcohol consumption, change in living situation, social isolation, family members expressed concerns or caregiver stress. Charts of patients ≥65 years of age were proactively reviewed to calculate the eFI score. All recruited patients with eFI > 0.12 were enrolled in the SCH pilot project.

### 2.3. Tools Used in the SCH

#### 2.3.1. Electronic Frailty Index (eFI)

Early stages of frailty can be challenging to identify, and therefore valid, reliable and acceptable tools should be used to augment clinical impressions. Examples of these tools have been described to help clinicians identify frailty [6], and different tools were tested within the SCH pilot site [23]. The eFI is a clinically validated tool that can identify older adults at risk of frailty using routinely collected primary care data [24]. It can be completed either in advance (by reviewing patient chart), or at the time of the clinic visit, with minimal disruption to clinic flow. The eFI consists of 36 deficits that include comorbidities, physical impairments, clinical signs, abnormal test values, and social circumstances [25]. In the pilot study, the eFI was calculated using manually retrieved structured and unstructured data from the electronic medical records (EMR) by a trained research coordinator. The eFI identifies people at high risk of adverse health outcomes and stratifies patients into fit (0–0.12), mild (0.13–0.24), moderate (0.25–0.36), and severe (>0.36) frailty [24].

#### 2.3.2. 4-Metre Gait Speed Test

A gait speed of 0.8 m/s or less (equivalent to taking longer than 5 s to walk 4 m) has high sensitivity (Sn = 0.99) and moderate specificity (Sp = 0.64) for the measurement of frailty [26]. Slow gait speeds are associated with falls, disability, institutionalization and death [27,28].

#### 2.3.3. Health-Related Quality of Life Measures (EQ-5D-5L and EQ-VAS)

The EQ-5D tool is a self-administered, standardized instrument consisting of a simple descriptive profile (i.e., EQ-5D-5L) that includes five domains (mobility, self-care, usual activities, pain/discomfort, and anxiety/depression) and a visual analogue scale (i.e., EQ-VAS) [29]. A value set for Canada was used to retrieve index values for each of 3125 possible health states, as defined by the EQ-5D-5L profile [30].

#### 2.3.4. Multi-Domain Assessment (MDA)

The MDA was designed for the primary care setting by SCH co-founders and is based on the principles of the Comprehensive Geriatric Assessment (CGA), the current gold standard of frailty identification and management [31]. Unlike the CGA that can be complex, labor-intensive, and expensive to operationalize [32], the MDA allows the assessment to be completed over multiple visits by a trained primary health care (PHC) provider such as a PHC nurse.

The MDA is a key information-gathering step that enables better understanding of the patient’s health context and provides the basis for managing frailty. The MDA accounts for the physical and mental health, social circumstances, functional capacity and the environment of a senior living with frailty, including screening for caregiver burnout.

Through its systematic evaluation, the MDA ensures that interventions for frailty management are appropriately aligned with the objective level of frailty [33] and mitigate against age discrimination. For example, patients with mild frailty may benefit from supported self-management, medication reviews and fall risk assessment and prevention; patients with moderate frailty—from care and support planning, in addition to the aforementioned interventions; and patients with severe frailty—from case management and end of life care. In the SCH, interventions are supported by the PCN interprofessional team (e.g., nurses, pharmacists, dietitians, mental health consultants and others) depending on the specific needs of the patient/caregivers and the expertise required. Linkages are made to local social and community services to meet holistic care needs.

#### 2.3.5. Person-Centered Care and Support Plan (CSP)

Care and support planning, or wellness planning, is the active discussion among patients, caregivers and health care providers as co-creators of the plan. Topics discussed include the goals and priorities of the patients and caregivers, chronic disease self-management, required supports, personal directives and preferences for end-of-life care, and other aspects relevant to the needs of a particular patient and their caregiver. Multicomponent interventions for management of frailty are based on the patient’s health needs, priorities and level of frailty, with involvement of various team members. Strategies may include physical activity, nutritional support, structured medication reviews and strengthening social networks. The COE physician was involved in care when there was diagnostic uncertainty, multiple complex issues, or challenging symptoms to control.

For improved efficiency in the SCH, templates were created within the EMR. This enabled a virtual space for team members to communicate, complete assessments, share clinical notes, and integrate the CSP. A copy of the agreed upon CSP and medication list are given to the patient and their designated caregiver for informational continuity.

### 2.4. Staff Training

The SCH leveraged their existing PCN’s multidisciplinary resources (e.g., nurses, pharmacists, dietitians, kinesiologists, social workers), and hired a geriatric assessment nurse (GAN) to support the education and facilitate collaboration within the team. Workshops and educational sessions with emphasis on concepts of frailty, case finding tools, MDA, care planning and interprofessional core competency were provided by the principal investigators to all clinic staff and health care providers at the start of the pilot project. Ongoing education and mentorship were provided by the COE and GAN. In addition clinic staff were educated about seniors’ community resources and supports in their area.

### 2.5. SCH Process of Care

Intervention steps of the process of care are outlined in Table 1. Included are tools used, providers involved, time to complete, method of documentation, and the resources and licenses required to conduct each step.

### 2.6. SCH Pilot Design

A quasi-experimental one group pretest-posttest design [34] was used to evaluate the SCH pilot data collected at the initial assessment (pre) and then again 12 months later (post). The study focused on measuring patient-oriented outcomes of health-related quality of life measures (EQ-5D-5L and EQ-VAS), as well as the 4-metre gait speed test. Results of the pharmacist-led structured medication reviews conducted as part of the SCH pilot have been published elsewhere [35].

The Health Research Ethics Board, University of Alberta, reviewed and approved the study protocol on 20 April 2016 (Study ID# Pro0062357). All participants provided written informed consent.

### 2.7. Analysis

Descriptive data was tabulated and presented as percentages. Demographic characteristics with normal distributions were presented as means and standard deviations, while medians and ranges were used to describe those characteristics that were not normally distributed. The change in each outcome from the initial assessment to the 12-month assessment was analyzed using Wilcoxon test as the data was not normally distributed. Statistical significance was established as a *p*-value less than 0.05. All data were analyzed using SPSS (Version 26.0) [36].

## 3. Results

The SCH model was piloted from July 2016–December 2018 in one clinic with six participating family physicians. Eighty-eight patients met the inclusion criteria and consented to participate in the study.

### 3.1. Study Population Description

#### 3.1.1. Population of Seniors Who Consented to Be in the SCH Model (N = 88)

In the sample of patients who consented to be in the SCH and satisfied the inclusion criteria, the mean age at enrolment was 81.1 (SD 7.5), most were females (60.2%), married (52.3%), and independently living at home (84.1%). Demographic characteristic of this sample can be seen in Table 2. The main reasons identified for attending the SCH were concerns of cognitive impairment/dementia (33.0%), and falls and decreased mobility (30.7%). The median number of chronic conditions reported per patient was 4.5 (with a range from 0 to 11). The most common chronic conditions were the following: hypertension (60, 68.2%), arthritis (36, 40.2%), and hyperlipidemia (26, 29.5%). Patients were taking a median of 8 medications that included both prescription and over-the-counter medications. According to the eFI, nearly 60% of the patients were moderately frail.

#### 3.1.2. Description of the Sub-Sample of Patients Who Took Part in the 12-Month Assessment (n = 38)

Thirty-eight of the 88 enrolled patients were assessed by SCH staff one year after their initial visit. The demographic characteristics of this sub-sample shown in Table 2 were similar to the initial assessment population, but with fewer patients living alone, and a higher median number of chronic conditions and medications per patient compared to the larger group of 88. However, the distribution of the eFI values was similar in both groups.

The reasons for patients not being assessed at 12 months (50) were as follows: patient could not be reached following multiple telephone attempts (2); collateral information from patients’ caregivers could not be arranged or obtained for assessment (3); patient died (9); patient and/or caregiver no longer interested in or felt they no longer needed SCH involvement (e.g., patients felt they had benefited from the first assessment, no longer needed SCH as linkages had been made to services in the community, were feeling well at the time of follow up, believed the assessments were too time consuming) (16); patient placed in long term care facility (7); and for unknown reasons (13).

#### 3.1.3. Health, Social & Community Services Linkages

A key feature of the SCH model is that clinic staff are aware of social and community services available to support patients and caregivers. SCH providers often provide information about these services to patients and their caregivers as part of setting their goals for care and jointly constructing a care plan. Examples of the types of linkages to health, social and community services are shown in Table 3. There was a median number of 2 linkages made per patient. While some patients did not require additional support through these linkages, other received up to 6 connections to other health providers or social and community services.

#### 3.1.4. Pre-Post EQ-5D-5L, EQ-VAS and 4-Metre Gait Speed Test Results

The SCH initial assessment (pre) and SCH 12-month assessment (post) measures of health quality of life (EQ-5D-5L and EQ-VAS) and the 4-metre gait speed test are presented in Table 4. Baseline and post-intervention 4-metre gait speed results, EQ-5D-5L and EQ-VAS scores did not differ. However, there was a shift in distribution within EQ-5D-5L domains of mobility, usual activities, pain/discomfort, and anxiety and depression towards Level 1 indicating better functioning.

Although 38 patients took part in the 12-month assessment, not all patients completed all parts. For example, assessments done at home visits frequently did not include the 4-metre gait speed test. As well, patients dependant on wheelchairs for mobility did not complete this test. Missing data in the self-administered EQ-5D-5L and EQ-VAS questionnaires was attributed to patient confusion about the meaning of some questions and other patients chose not to complete the questionnaire.

## 4. Discussion

The SCH is a holistic, integrated model of care for community dwelling older adults with frailty. This model is well aligned with WHO publications addressing healthy ageing and integrated care for older people [11,12,37] as well as international recommendations for effective integrated care models [13,16,17,19,38,39]. The SCH model for primary care providers has evolved over time to its current state, beginning slowly with assembling an interprofessional primary care team and training the team in geriatric competencies, and awareness of social and community resources and services. This paper has outlined the multiple intervention steps of the SCH with descriptions of each step including the providers involved, estimated times for completion, how to document and communicate findings, as well as tools, resources, and associated licenses required for each step.

Evaluation of the SCH pilot presented in this study included the 4-metre gait speed test and the patient-oriented outcome measure, EQ-5D-5L which include the five domains of mobility, self-care, usual activities, pain/discomfort, and anxiety/depression as well as the visual analogue scale (EQ-VAS). Choosing patient-oriented outcomes supports the work by the International Consortium for Health Outcomes Measurement to use outcomes that are meaningful to patients so that they receive care aligned with the outcomes they value [40]. SCH pilot outcomes were measured at the time of the initial assessment (pre) and again at the 12-month assessment (post) to test the effect of the SCH as an intervention. Despite small improvements observed in the EQ-5D-5L domains of mobility, usual activities, pain/discomfort, and anxiety and depression, the analysis indicated no statistically significant changes from the initial assessment to the 12-month assessment.

It is likely that the lack of change observed is related to the relatively short, follow-up duration of 12 months. This finding is consistent with recommendations made by others studying the frail older population [38,41,42]. In particular, inadequate study durations in randomized controlled trials have been identified as contributing to intervention groups showing little or no change in outcomes [38]. The PRISMA integrated care model established throughout regions of Quebec, Canada did not demonstrate improved patient outcomes, health utility and cost effectiveness until it was in place for four years [41]. These evaluations indicated the PRISMA model site versus comparator sites resulted in less functional decline, reduced emergency room visits, fewer unmet needs, and increased client satisfaction and empowerment [41].

Additionally, larger sample sizes are also needed to evaluate the impact of integrated care models for older adults with spectrum of frailty. Smaller participant numbers tend to include a range of frailty levels (mild to severe) as observed in the SCH pilot. This heterogeneity is problematic in teasing out whether patient outcomes were affected by the integrated care model within categories of frailty [38]. For example, changes in outcomes are less likely to be observed in patients with mild frailty in the short term or in those with severe frailty. It has been suggested that some integrated care model studies were able to improve functional disabilities but did so by selecting patients most likely to benefit from the intervention [43]. In reference to integrated care models that fail to show effects on health-related quality of life indicators such as the EQ-5D, Ekdahl et al. posited that this lack of effect may reflect patients’ poor health status initially, low health related quality of life, and therefore, improvements in these areas are likely limited [44]. Larger patient numbers may allow for sub-group analysis of the results according to patients’ frailty levels. This may also help direct interventions to groups who would benefit the most from the interventions.

Another feature of successful integrated care models for older persons with frailty is the consistent and large-scale support from regional, provincial and national governments [11,19,41,45,46]. Interestingly, researchers from countries with highly developed primary care programs have reported difficulties observing effects in their intervention groups. They suggest the reason is because control groups are already receiving high quality care as their “usual care” [38,42,44]. In Canada where health care is under the provincial domain, expansion of integrated care models in provinces such as Quebec and British Columbia has occurred through regional and provincial government support [41,47]. As well, in Quebec having social and health care services delivered by one Ministry has facilitated the integration of health, community and social services [41]. These factors enabled the development and growth of the PRISMA model which is case manager led and delivered as an external program independent of the patient’s primary care provider [48]. However, in most other Canadian provinces including Alberta, large-scale government support for integrated care has not been established. An advantage of the SCH framework is that it can be implemented as a standalone model within a primary care clinic, utilizing existing resources to ensure its long-term sustainability. Therefore, SCH fulfills a need for primary care providers who wish to establish an integrated model of care for adults with frailty in their practice, but may not have the benefit of broad regional funding, resources and support to do so.

### 4.1. Limitations

There are limitations to the pilot study evaluation. These include a small sample size and short duration of follow-up previously discussed as well as the low retention rates of the study population. Measures were taken to improve retention rates (e.g., information provided by nurse caseworker about the study processes and appointment reminders) and were consistent with those recommended in the literature [49]. However, less than half of the enrolled participants completed their 12-month assessment. In addition, the pilot study took place in one urban, academic, teaching primary care clinic and therefore the pilot results may not be generalizable to other non-academic or rural primary care clinics.

### 4.2. Lessons Learned

To provide actionable information to those planning to implement a similar integrated care model, we mapped the SCH team’s experience and recommendations to the five components of ADKAR model (Awareness-Desire-Knowledge-Ability-Reinforcement) [22].

AWARENESS is about establishing the need for change. This is a critical step because clinics are already time poor and some changes might be perceived as adding to an already stretched primary care system where providers’ “buckets are full.” SCH clinical leads (MA, SK) invested the time to listen to all members of the clinic, including health professionals, quality improvement personnel, and receptionists. Staff shared stories of their frustrations with the current state of health. Many had cared for patients in the clinic or personally experienced the impacts of frailty, such as patient/caregiver struggles with navigating the system, adverse effects of polypharmacy, not knowing how to help their complex seniors presenting to clinic, and the long wait times for referrals. The goals of SCH were shared to demonstrate how the model would not add to, but rather support their ability to provide holistic senior’s care and address the frustrations they had voiced. The SCH model of capacity building in primary care clinics was also aligned with priorities identified by the provincial health authority in Alberta to improve care for vulnerable seniors whose population was projected to rise over the next two decades, and seniors with frailty were noted among the top users of healthcare.

DESIRE is about individuals’ willingness to support and take part in a change. Staff participated in open dialogues about how they could make an impact within their role, and how the SCH would benefit both patient care and support their professional goals and development. It also involved stakeholder engagement, including that of the Primary Care Network, so that the staff were supported and that potential barriers to change were addressed. It was evident from a review of provincial health service data that the Edmonton Oliver PCN cared for one of the largest populations of complex older adults, thus reinforcing their support for the SCH innovation and the justification for reallocation of resources. Moreover, throughout the implementation process, team feedback and recommendations were not only sought but acted upon. This further validated the importance of each team member’s contribution and catalyzed collaborative practice.

KNOWLEDGE is about ensuring that all those involved in care for seniors living with frailty understand their roles and responsibilities, that processes are in place to support them, and that individuals receive appropriate training. Clinic staff were invited to workshops and educational sessions to learn about the spectrum of frailty, proposed SCH process of care, and their role in the SCH implementation. A geriatric nurse specialist was hired to support the education and facilitate collaboration within the team. All team members were encouraged to approach the geriatric nurse specialist and COE for one-on-one teaching (e.g., case reviews, consultations). Pharmacists were supported to complete advanced geriatric training and achieved pharmacy board certified geriatrics credentials.

ABILITY is about taking the knowledge gained and applying it in the real-world environment. It was important in the SCH to “keep a finger on the pulse”, to mitigate any negative attitudes towards challenging status quo in the primary care setting. During the implementation of the SCH, weekly “team huddles” were held. This enabled each person to quicky share what was going well, what was not going well, and suggestions for change. This enabled timely adjustments and changes in the processes to better meet provider, patient and caregiver needs. For example, the primary health care nurses noted that some of the frailty identification tools disrupted clinic flow or were difficult for patients to complete. To address this, different case finding tools were considered, and the eFI was selected as it was found to be the most convenient and efficient tool to use in the clinic. Key patients and caregivers who readily volunteered their perspectives about health care were also approached for feedback and engaged in all stages of the SCH, including design, implementation, and evaluation.

REINFORCEMENT involves sustaining change through actions or events that strengthen, emphasize or highlight the value of the changes made. Stories of patients and caregivers experiencing improved health and high satisfaction with care were shared by all clinic staff to encourage staff and help reinforce their efforts. Over time, health professionals reported a better grasp of the health status and needs of their older adult patients, as well as being supported in addressing those uncovered biopsychosocial needs. Having an evaluation framework helped to guide meaningful measurement (i.e., in a busy primary care clinic what outcomes matter the most?). However, in reflection of the SCH pilot, this data could have been shared more frequently and at regular intervals with the team to further reinforce the value and need to measure consistently. Another strong reinforcement was that the success of the pilot resulted in the expansion of the SCH to serve additional clinics within the PCN.

### 4.3. Future Research Direction

As more primary care practices adopt a structured approach for frailty assessment and management, such as the SCH model, more robust data can be consistently collected from which larger scale studies designed to demonstrate the impact of integrated models. In addition, with an emphasis on upstream interventions in earlier stages of frailty, further investigations of patient-oriented outcomes, impacts on quality of life measures, and cost-effectiveness, should be the focus, rather than traditional outcomes of institutionalization and mortality associated with late stages of frailty. Further, in the setting of a primary care integrated model for complex community-dwelling older adults, where team members are not co-located, an opportunity exists to study the development and impact of interprofessional competencies and shared mental modeling.

## 5. Conclusions

The SCH is a grassroots, innovative integrated model of care that outlines a framework for upstream primary care assessments and interventions for older adults at risk of frailty. With creative mobilization of available resources and services any primary healthcare team can implement the SCH within their practice. The lack of statistically significant improvements in patient-oriented outcomes shown in the 12-month evaluation of the SCH pilot underlines the need for future studies with larger patient numbers and longer durations of follow-up to demonstrate impacts of integrated models in the primary care setting.

## Figures and Tables

**Table 1 geriatrics-06-00018-t001:** SCH process of care.

Intervention Steps	Tools Used, Providers Involved, Time to Complete, Method of Documentation, Tools, Resources and Licenses Required
1. Frailty Identification	**Tool used:** Electronic frailty index (eFI). Patients at risk of frailty (eFI > 0.12) go on to Step 2. **Providers Involved:** Nurse, PCP ^1^ and other clinical team members **Time to complete eFI:** Approx. 5 min. (Depends on chart documentation available and assessor’s knowledge of patient.) **Method of documentation:** EMR ^2^ template (36 deficit checkbox and automatic eFI score calculation), but can also be done on paper & entered into patient chart **Resources/Licenses:** eFI publicly available
2. Frailty Assessment	**Tool used:** Multi-Domain Assessment (MDA). Covers medical/physical, social, psychological, cognitive, environmental and functional domains. **Providers Involved:** trained SCH clinician with multidisciplinary team support. Pharmacist trained in geriatrics completes medication review. **Time to complete assessment:** 1–2 h, can be done over multiple visits. **Method of documentation:** EMR or paper version. **Resources:** MDA template built into EMR. Additional tools needed for assessment (e.g., vitals, cognitive testing, portable hearing amplifier). **Licenses:** Assessment tools–MDA template (Available upon request). 4-metre gait test (Publicly available in English), EQ-5D/VAS available in English/ French. (Permission required from EuroQol Office.)
3. Frailty Management	**Tool used:** Care and Support Plan (CSP) template (data automatically pulled from MDA template). **Providers Involved:** SCH clinician with multidisciplinary team support, PCP, patients/caregivers, +/− COE ^3^ (e.g., diagnostic uncertainty or multiple complex issues/challenging symptoms to control); SCH clinician follows up as needed with patient/caregiver (by telephone or in-clinic visit) and hands-off to appropriate case manager (e.g., Home Care, PCP, PHC ^4^ nurse). **Time to complete:** Charting MDA (summarizing issues/ interventions, 1–2 h), CSP (~20 min). **Method of documentation:** EMR or paper version. **Resources:** Team members to support interventions & link patients to social/community resources. **Licensing:** CSP template (Available upon request).

^1^ primary care physician, ^2^ electronic medical record, ^3^ Care of the Elderly, ^4^ primary health care.

**Table 2 geriatrics-06-00018-t002:** Characteristics of patients at initial SCH and at 12-month assessment.

Characteristic	Initial (N = 88) N (%)	12 Month (n = 38) N (%)
Female sex	53 (60.2)	21 (55.3)
Age in years (mean, SD)	81.1, 7.5	80.4, 6.4
Marital status:		
Single	8 (9.1)	3 (7.9)
Married/common-law partner	46 (52.3)	23 (60.5)
Divorced/separated	5 (5.7)	1 (2.6)
Widowed	28 (31.8)	11 (28.9)
Unknown	1 (1.1)	0
Reason for assessment:		
Caregiver burden	10 (11.4)	5 (13.2)
Chronic pain	16 (18.2)	3 (7.9)
Depression	15 (17.0)	7 (18.4)
Cognitive impairment/dementia	29 (33.0)	12 (31.6)
Failure to thrive	2 (2.3)	1 (2.6)
Falls and decreased mobility	27 (30.7)	11 (28.9)
Home support	2 (2.3)	0
Medication review or polypharmacy	10 (11.4)	3 (7.9)
Medically complex	10 (11.4)	5 (13.2)
Living alone	30 (34.1)	6 (15.8)
Accommodation:		
Independent home living	74 (84.1)	32 (84.2)
Private supportive living	11 (12.5)	5 (13.2)
Designated supportive living	2 (2.3)	1 (2.6)
Education:		
No formal education	1 (1.1)	0
Primary (K-Grade 9)	16 (18.2)	7 (18.4)
Secondary (Grade 10–12)	39 (44.3)	19 (50)
Post-secondary	31 (35.2)	12 (31.6)
Unknown	1 (1.1)	0
Chronic Conditions:		
Hypertension	60 (68.2)	24 (63.1)
Arthritis	36 (40.9)	17 (44.7)
Hyperlipidemia	26 (29.5)	14 (36.8)
Atrial fibrillation	21 (23.9)	9 (23.7)
Chronic obstructive pulmonary disease	21 (23.9)	8 (21.0)
Chronic conditions/patient, M, Range	4.5, 0–11	5.0, 2–8
Medications/patient, M, Range	8.0, 1–19	8.5, 2–19
eFI, Me, SD	0.30, 0.10	0.30, 0.10
mild (0.13–0.24)	15 (18.3)	8 (21.0)
moderate (0.25–0.36)	49 (59.8)	24 (63.2)
severe (<0.36)	18 (22.0)	6 (15.8)

**Table 3 geriatrics-06-00018-t003:** Health, social & community services linkages made by SCH clinic staff.

Types of Linkages Made (after Initial SCH Assessment Visit)	N (%)
Primary Care Network (PCN) team members involved in care:	
Dietitian	7 (3.7%)
PCN pharmacist	62 (32.5%)
PCN social worker	1 (0.5%)
PCN geriatric assessment nurse	41 (21.5%)
Other PCN team members (e.g., mental health nurse, kinesiologist)	18 (9.4%)
Care of the Elderly Physician	13 (6.8%)
PCN programs (e.g., mental health, nutrition, active living & lifestyle programs/group learning courses)	4 (2.1%)
Medical specialties (e.g., Neurology, Neuropsychiatry, Cardiology, Geriatric Psychiatry)	2 (1.0%)
Day Programs (e.g., ElderCare Edmonton Day Programs; CHOICE Day program)	4 (2.1%)
Home Care Services (e.g., Homecare Alberta Health Services-for ongoing support, Home Safety Assessment-short term support for occupational therapy and equipment)	17 (8.9%)
Rehab Programs (e.g., Better Balance/Outpatient Rehab program)	19 (9.9%)
Other linkages made (e.g., Alzheimer Society, SAGE Seniors’ Association, Caregivers Alberta)	3 (1.6%)
Total number of linkages	191
Number of linkages per patient (n = 88), M, Range	2.0, 0–6

**Table 4 geriatrics-06-00018-t004:** Changes in EQ-5D-5L, EQVAS and 4-metre gait speed test from initial SCH to 12-month assessment.

Outcome Measure	Initial	12-Month	P
4-metre gait speed test: M (N, Range)	4.83 (36, 3.1–11.7)	4.83 (30, 3.1–11.6)	0.600
EQ-5D-5L index: M (N, Range)	0.846 (37, −0.270–0.949)	0.817 (36, −0.066–0.949)	0.926
EQ-5D-5L mobility: M (N, Range)	2.00 (36, 1–5)	2.00 (36, 1–5)	0.297
Level 1	11 (28.9)	10 (26.3)	
Level 2	10 (26.3)	14 (36.8)	
Level 3	8 (21.0)	9 (23.7)	
Level 4	6 (15.8)	1 (2.6)	
Level 5	1 (2.6)	2 (5.3)	
Unknown	2 (5.3)	2 (5.3)	
EQ-5D-5L self-care: M (N, Range)	1.00 (36, 1–5)	1.00 (36, 1–5)	0.564
Level 1	26 (68.4)	26 (68.4)	
Level 2	6 (15.8)	4 (10.5)	
Level 3	2 (5.3)	4 (10.5)	
Level 4	0	0	
Level 5	2 (5.3)	2 (5.3)	
Unknown	2 (5.3)	2 (5.3)	
EQ-5D-5L usual activities: M (N, Range)	2.00 (36, 1–5)	1.00 (36, 1–5)	0.181
Level 1	13 (34.2)	19 (50.0)	
Level 2	12 (31.6)	9 (23.7)	
Level 3	9 (23.7)	5 (13.2)	
Level 4	0	1 (2.6)	
Level 5	2 (5.3)	2 (5.3)	
Unknown	2 (5.3)	2 (5.3)	
EQ-5D-5L pain/discomfort: M (N, Range)	2.00 (36, 1–4)	2.00 (36, 1–4)	0.559
Level 1	11 (28.9)	12 (31.6)	
Level 2	8 (21.0)	9 (23.7)	
Level 3	12 (31.6)	11 (28.9)	
Level 4	5 (13.2)	4 (10.5)	
Level 5	0	0	
Unknown	2 (5.3)	2 (5.3)	
EQ-5D-5L anxiety/depression: M (N, Range)	1.00 (36, 1–4)	1.00 (36, 1–4)	0.412
Level 1	20 (52.6)	21 (55.3)	
Level 2	6 (15.8)	6 (15.8)	
Level 3	7 (18.4)	8 (21.0)	
Level 4	3 (7.9)	1 (2.6)	
Level 5	0	0	
Unknown	2 (5.3)	2 (5.3)	
EQ-VAS, M (N, Range)	80.0 (38, 25–100)	80.0 (34, 10–100)	0.565

## Data Availability

The data presented in this study are available on request from the corresponding author. The data are not publicly available as subjects did not agree to this. However, subjects agreed to have their data used in future research pending approval from the Health Research Ethics Board, University of Alberta.

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
