# Peer review of "The Seniors’ Community Hub: An Integrated Model of Care for the Identification and Management of Frailty in Primary Care"

_geriatrics, 2021, doi:10.3390/geriatrics6010018_

Round 1

Reviewer 1 Report

Frailty and the provision of integrated, as well as its impact on older adults’ health outcomes are topics of high relevance. 

The model presented in the manuscript is a well-designed intervention for integrate care. However, the evaluation provided in this study could be improved. For instance, by conducting several follow-up assessements (e.g. at 6-months; 12-months and 24 months) and also by recruiting a larger sample. 

The study is methodologically correct and the results are well-described, even though most of them are not statistically significant. 

Few minor suggestions for improvement: 

  • In the results section, more details on why some participants did not complete all parts of the 12-months follow-up evaluation should be provided. 
  • In the analysis section the SPSS version is not included. 
  • Tables' format should be reviewed. 

Author Response

Please see attached word document (Geriatrics-1105453_Response to Reviewer 1) for our responses to Reviewer 1.

Reviewer 2 Report

This paper describes the Seniors' Community Hub, a model of care to identify and support frail older adults through the primary care system. The model is described clearly, and while the pilot participants did not show significant improvements after follow-up testing, it is reasonable to assume from the approach that, in principle, this model has great potential for improving community health.

However, the paper could benefit from additional content describing the process of setting up the SCH, the challenges observed over the course of the pilot, and the mechanisms/approach used to engage the health care providers as well as members of the community. The "Lessons Learned" section is valuable, but as none of those areas were mentioned previously in the paper, they feel like they are not connected to the prior discussion. Describing how the lessons learned emerged from the presented methods and data, or providing more detail about the 'start-up' experience, would make this paper much more comprehensive. As it stands, it does not sufficiently describe how another group might take these learnings and implement their own version of the SCH, and since the participants did not show any significant improvements, this omission diminishes the overall impact of the paper.

Author Response

Please refer to the uploaded word document (Geriatrics-1105453_Response to Reviewer 2) for our responses to Reviewer 2
